# Transient superconductivity in three-dimensional Hubbard systems by combining matrix product states and self-consistent mean-field theory

S. Marten[1], G. Bollmark[2], T. Köhler[2], S.R. Manmana[1], A. Kantian[2,3]

**1** Institut für Theoretische Physik, Georg-August-Universität Göttingen, 37077 Göttingen, Germany **2** Department of Physics and Astronomy, Uppsala University, Box 516, S-751 20 Uppsala, Sweden **3** SUPA, Institute of Photonics and Quantum Sciences, Heriot-Watt University, Edinburgh EH14 4AS, United Kingdom

## Abstract

We combine matrix-product-state (MPS) and mean-field (MF) methods to model the real-time evolution of a three-dimensional (3D) extended Hubbard system formed from one-dimensional (1D) chains arrayed in parallel with weak coupling in-between them. This approach allows us to treat much larger 3D systems of correlated fermions out-of-equilibrium over a much more extended real-time domain than previous numerical approaches. We deploy this technique to study the evolution of the system as its parameters are tuned from a charge-density wave phase into the superconducting regime, which allows us to investigate the formation of transient non-equilibrium superconductivity. In our ansatz, we use MPS solutions for chains as input for a self-consistent time-dependent MF scheme. In this way, the 3D problem is mapped onto an effective 1D Hamiltonian that allows us to use the MPS efficiently to perform the time evolution, and to measure the BCS order parameter as a function of time. Our results confirm previous findings for purely 1D systems that for such a scenario a transient superconducting state can occur.

# 1   Introduction

Superconductivity (SC) has remained a phenomenon of great interest to researchers ever since its discovery in 1911 by H. K. Onnes. Explaining SC in metals at low-temperature equilibrium was already a challenge, taking more than 40 years until the Bardeen-Cooper-Schrieffer (BCS) framework could explain it via a suitable mean-field (MF) theory. In the 1980s, SC at high critical temperatures $T_c$ [1–4] was discovered, which seemed not to be described by BCS theory. In fact, its theoretical description presents a still-ongoing challenge. It is believed that strongly correlated electron motion is the underlying reason for this type of SC state. Many-body models such as the Hubbard- [5–10] or the $t$-$J$-model [4, 10–13] have been investigated to study this question. In more recent developments, experiments claim to have detected transient[1], light-induced SC states after pushing layered high-$T_c$ materials out-of-equilibrium in pump-probe setups, even above the equilibrium $T_c$ [14–18]. One interpretation of these results is via the concept of pre-formed pairs, namely that above $T_c$ up- and down-spin electrons are still paired, up to the so-called pseudogap (PG) temperature $T_{\mathrm{PG}}$. In this view, at temperature $T$ with $T_c < T < T_{\mathrm{PG}}$, the system cannot achieve SC order in equilibrium because the pre-formed pairs within each layer lack inter-layer phase coherence. Thus, the pump-pulse is deliberately designed to increase inter-layer coupling, and results in the observation of a seemingly SC response in the optical conductivity, but only as long as $T < T_{\mathrm{PG}}$ [16, 17].

There are two possible objections to this particular interpretation of the experimental literature: **(1)** The reliance of experiments on the time-dependent optical conductivity as a probe for nonequilibrium SC has been questioned. Paeckel et al. [19] recently showed that this measure lacks specificity for SC order and proposed alternative measurements, which would be better suited to detecting the onset of the SC state in the dynamically evolving system. The setup studied in that article consists of a quench performed on a one-dimensional (1D) extended Hubbard system at $T = 0$ using a matrix-product-state (MPS) description. This MPS approach, while unbiased and highly accurate, however was so far effectively restricted to 1D systems, especially when treating out-of-equilibrium dynamics. The question is thus if the findings of Paeckel et al. are specific to 1D, with its strong quantum and thermal fluctuations, or whether they also apply to higher dimensional systems. **(2)** This leads to the second objection: quantitative or even qualitative theoretical understanding of the solid state high-$T_c$ materials evolving under a pump-pulse probe, or even a generalization of the setup of Paeckel et al. to a three-dimensional (3D) system, has been lacking (even understanding the high-$T_c$ materials in equilibrium still presents an enormous challenge). That is, no one has been able to show even in principle whether the proposed scenario for interpreting many solid-state experiments – initially phase-incoherent pairs of fermions dynamically acquiring macroscopic coherence as inter-layer coupling is rapidly increased – is theoretically possible, and, if so, under which conditions and on which time scales. While some numerical work has been carried out towards this, [19–21], many basic questions about the mechanisms that could lead to dynamically induced SC remain open. To a substantial degree, this is due to the significant challenge of accurately capturing those system sizes and time scales required to observe any emerging SC states.

The central question then becomes which class of algorithms has any chance of reaching those regimes, thereby delivering the currently-lacking theoretical insight into how dynamically induced SC develops and which microscopic conditions would be fundamentally required for this. On their own, even in 1D, MPS methods may require exponentially increasing resources as simulation time grows in order to maintain a set accuracy. This is due to the strong growth in bipartite entanglement in these systems with time: for

---

[1]termed "transient" because they have short lifetimes when induced above equilibrium $T_c$

MPS approaches to be efficient, this entanglement should not be too large. Furthermore, already for equilibrium calculations long-range interactions, which are needed to represent two-dimensional (2D) and 3D systems in 1D, increase the entanglement dramatically. Hence, the time evolution of generic 2D and 3D systems are entirely out of reach for any brute-force MPS-based approach, as they could not capture even the initial equilibrium state.

For such higher-dimensional systems, real-time non-equilibrium dynamical mean-field theory (DMFT) could be a powerful alternative approach [22, 23]. For this technique, one or a few lattice sites – the impurity or, respectively, the cluster – are retained explicitly, including all interactions of the original, infinitely-large lattice. In DMFT, the effect of this remainder-lattice on the cluster is mimicked via a free-electron bath that is coupling to it. The parameters of this bath are fixed via self-consistency conditions. Solving these cluster-bath systems within this self-consistency constraint is typically achieved by applying quantum Monte Carlo (QMC) techniques in the real-¸ time domain. These techniques suffer from a strong sign-problem, i.e., their numerical error grows exponentially as the cluster-size and the real-time domain, over which the simulation runs, are increased. In practice, a few sites and time scales on the order of the electron tunneling are accessible. Alternatively, MPS solvers can be used within such real-time non-equilibrium DMFT; however, due to the long-range tunneling in these systems between bath and cluster sites, and the strong growth of entanglement with time, these will also be limited to a few sites and short times.

This leads us to the scope of the present paper: with current methods it seems practically impossible to perform meaningful simulations of dynamically-induced SC in a 3D system. For MPS methods, the growth of entanglement with system size and simulation time is immediately prohibitive; and for non-equilibrium real-time DMFT, the large clusters and long times required to resolve the onset of a potentially weak SC order appear to be out of reach.

However, as we demonstrate in the following, it is possible to make a specific category of 3D fermionic systems amenable to real-time evolution via MPS techniques using a static MF ansatz, by exploiting certain gaps in the excitation spectrum of these cases. In this way, it is possible to capture strong correlations by using a MPS, and treat the full 3D system more accurately than by applying a pure MF treatment. We show explicitly that our approach can describe the dynamical emergence of SC from a state that is not SC to begin with in a PG-like system, i.e., a system that has singlet-pairing of spinful fermions but with no initial phase coherence of these pairs. In this, our method can address exactly the setup thought to be at the heart of the solid state experiments, and for which we present here the first efficient many-body numerics. Several of the authors have previously developed related approaches for systems in equilibrium [24], reproducing physical behavior correctly at zero and finite temperature compared to appropriate QMC simulations [25, 26], with the overestimation of SC properties due to the MF-component of our technique a constant one, and modest at that in 3D systems. In these approaches, weakly coupled chains or ladders are stacked up into 3D cubic systems, which thus have anisotropic tunneling — much stronger inside the 1D systems than in-between them in the two orthogonal directions. For the case of fermions, the MF approximation can be introduced if each of the constituent 1D systems has a gapped energy sector, such as a spin gap, and thus single-fermion tunneling in-between 1D systems is suppressed in this weak-coupling regime [25]. Just as for the equilibrium case [25], it is this crucial ingredient that allows us to perform real-time evolution for a much higher number of correlated sites than non-equilibrium real-time DMFT, as well as extending the real-time domain enough to perform a meaningful simulation of the dynamically-induced SC in a 3D system. Within

this well-behaved domain, we apply our real-time MPS+MF technique to study the time evolution of the BCS order parameter after a fast ramp of the system from an insulating starting state into a parameter regime where the system would be SC in equilibrium. As a consequence, we observe the onset of a non-equilibrium SC state.

We note that the present work is focused purely on the algorithmic challenge of numerically studying such state, which entails simulating many-body non-equilibrium systems of fermions for system sizes and time scales that were previously inaccessible. As to how the interrelated questions of validating our approach with alternative methods and applying it to concrete experiments could be answered, we provide an outlook to that in the Conclusions. Further, as shown in Sec. 3 and Fig. 2, the basic algorithm is completely generic, i.e., it is not predicated on the dynamics of the system being of a specific type. Thus, our approach can be immediately generalized to quasi 1D systems with, e.g., explicit periodic driving (i.e., Floquet-type dynamics [27,28]), or non-unitary dynamics as encountered in open quantum systems (quantum trajectories [29] or full master-equation [30] for Markovian baths, hierarchy of pure states (HOPS) [31] or projected purified MPS (ppMPS) enabled quantum trajectories [32] for non-Markovian ones); MPS-approaches have extensive track records in all these domains, but each of them would represent separate projects of their own and are thus not the subject of the present work.

The paper is structured as follows: in Sec. 2, we recapitulate the MF ansatz for weakly coupled Hubbard chains used in equilibrium, developed originally in [25]. In Sec. 3, we introduce the extension to a self-consistent time-dependent MPS+MF scheme to study the time evolution of a 3D extended Hubbard system, which consists of weakly coupled chains. In Sec. 4, we present our results for the BCS order parameter and a detailed discussion of the convergence behavior of the method when treating 3D arrays formed from chains, each up to $L = 30$ lattice sites long. The time evolution of the SC order parameter shows indeed that in both finite systems as well as the thermodynamic limit a transient SC state can be entered. We further analyze the dependence of our results on the parameters of the simulations. In Sec. 5 we conclude and provide an outlook as to how the technique established in the present work could be validated and applied to experiments. The appendices discuss further details on the method at equilibrium, as well as further details of the simulations out-of-equilibrium.

## 2    Mapping of the 3D system onto a 1D self-consistent chain

As we aim to describe a 3D model system with a method that is mainly suitable for 1D, namely MPS, we first need to identify a class of 3D models amenable to mapping onto an effective 1D description. Following the work of Bollmark et al. [25,26], we focus on 3D systems constructed out of gapped 1D fermions. We arrange these 1D systems, which extend in the $\hat{x}$-direction, in parallel into a square array in the $\hat{y} - \hat{z}$-plane, forming effectively a cubic lattice. We choose fermion tunneling to be anisotropic in this lattice, denoted by $t_\perp$ in the $\hat{y}$- and $\hat{z}$-directions. Adapting from Bollmark et al. [25], we choose an extended Hubbard chain as the 1D building block. The Hamiltonian constructed in this manner is illustrated in Fig. 1 and is given by

$$\hat{H} = \hat{H}_0 + t_\perp \hat{H}_\perp \, , \tag{1}$$

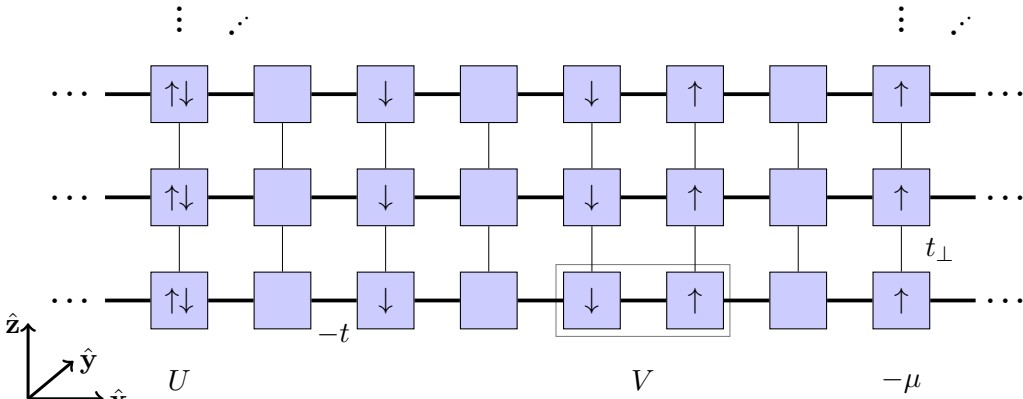

Figure 1: Two-dimensional cross-section of the three dimensional model. For the sake of clarity, the 3D extension of the system out of the plane is not shown here. Each box denotes a lattice site. The sites are coupled into chains in $\hat{\mathbf{x}}$-direction, as indicated by the thick lines between the boxes. Furthermore, all chains are weakly coupled by the transverse hopping $t_\perp$. This way, we obtain an extension in $\hat{\mathbf{y}}$ and $\hat{\mathbf{z}}$-direction.

with

$$\hat{H}_0 = -t \sum_{n=1}^{L-1} \sum_{\sigma \in \{\uparrow,\downarrow\}} \sum_{\{\mathbf{R}_i\}} \left( \hat{c}^\dagger_{n+1,\mathbf{R}_i,\sigma} \hat{c}_{n,\mathbf{R}_i,\sigma} + \text{h.c.} \right) - \mu \sum_{n=1}^{L} \sum_{\sigma \in \{\uparrow,\downarrow\}} \sum_{\{\mathbf{R}_i\}} \hat{n}_{n,\mathbf{R}_i,\sigma} \tag{2}$$

$$+ U \sum_{n=1}^{L} \sum_{\{\mathbf{R}_i\}} \hat{n}_{n,\mathbf{R}_i,\uparrow} \hat{n}_{n,\mathbf{R}_i,\downarrow} + V \sum_{n=1}^{L-1} \sum_{\sigma,\sigma' \in \{\uparrow,\downarrow\}} \sum_{\{\mathbf{R}_i\}} \hat{n}_{n+1,\mathbf{R}_i,\sigma} \hat{n}_{n,\mathbf{R}_i,\sigma'} \,, \tag{3}$$

and

$$\hat{H}_\perp = -\sum_{n=1}^{L} \sum_{\sigma \in \{\uparrow,\downarrow\}} \sum_{\{\mathbf{R}_i\}} \sum_{\hat{\mathbf{a}} \in \{\hat{\mathbf{y}},\hat{\mathbf{z}}\}} \left( \hat{c}^\dagger_{n,\mathbf{R}_i+\hat{\mathbf{a}},\sigma} \hat{c}_{n,\mathbf{R}_i,\sigma} + \text{h.c.} \right) \,. \tag{4}$$

Here, $\hat{c}^\dagger_{n,\mathbf{R}_i,\sigma}$ and $\hat{c}_{n,\mathbf{R}_i,\sigma}$ denote the fermionic creation and annihilation operators on site $n$ and for spin $\sigma$ on a chain that is labeled by the 2D vector $\mathbf{R}_i$ in the $\hat{\mathbf{y}} - \hat{\mathbf{z}}$-plane. They obey the anticommutation relations $\{\hat{c}_i, \hat{c}^\dagger_j\} \equiv \hat{c}_i \hat{c}^\dagger_j + \hat{c}^\dagger_j \hat{c}_i = \delta_{ij}$ and $\{\hat{c}_i, \hat{c}_j\} = \{\hat{c}^\dagger_i, \hat{c}^\dagger_j\} = 0$. The indices $i$ and $j$ stand for different combinations of $n, \mathbf{R}_i$, and $\sigma$. The operator $\hat{n}_{n,\mathbf{R}_i,\sigma} = \hat{c}^\dagger_{n,\mathbf{R}_i,\sigma} \hat{c}_{n,\mathbf{R}_i,\sigma}$ is the particle number operator for the corresponding site, chain, and spin. We use open boundary conditions and include a term for the chemical potential $\mu$. The latter allows us to control the number of particles in the system.

The only non-1D term is the transverse hopping $\hat{H}_\perp$. We are able to eliminate the beyond-1D nature of this term through a combination of perturbation theory on the transverse hopping and a MF decoupling of adjacent 1D systems. In the following we briefly recap the key steps, a detailed derivation of this approach can be found in the publication of Bollmark et al. [25].

Since we are interested in a model system for SC, we specify $U < 0$ in the chain-Hamiltonian Eq. (3). This negative-$U$ term gives rise to pairing of opposite-spin fermions already in isolated systems at $t_\perp = 0$. This is expressed by the finite spin gap $\Delta E_s$ and a

finite pairing energy $\Delta E_p$ of these isolated chains, defined as follows:

$$\Delta E_s(N) \equiv \mathcal{E}_0(1, N) - \mathcal{E}_0(0, N), \tag{5}$$

$$\Delta E_p(N) \equiv 2\mathcal{E}_0\left(\frac{1}{2}, N+1\right) - \mathcal{E}_0(0, N) - \mathcal{E}_0(0, N+2) . \tag{6}$$

Here, $\mathcal{E}_0(S_z, N)$ denotes the ground-state energy of Hamiltonian $\hat{H}_0$ for a single chain-index at total spin $S_z$ and total number of fermions $N$. Thus, $\Delta E_s$ and $\Delta E_p$ represent the minimal energy required for flipping a spin inside a chain and for breaking up a pair on a chain by moving one constituent to another chain in the full 3D system, respectively. From the definitions, it is easy to see that $\Delta E_s \leq \Delta E_p$, and for our specific choice of 1D systems $\Delta E_s = \Delta E_p$. As outlined in the following, $\Delta E_p$ becomes important in the actual numerical routine, directly entering the effective Hamiltonian Eq. (13). In practice, we can determine $\Delta E_p$ from a single chain via an extrapolation in the system size $L \to \infty$.

To carry out the second-order perturbation theory in $\hat{H}_\perp$ – specifically in $t_\perp/\Delta E_p$ – we follow [33]. We sort the eigenenergies $E_{i,\alpha}$ of $\hat{H}_0$ ($\hat{H}_0 |i, \alpha\rangle = E_{i,\alpha} |i, \alpha\rangle$) into manifolds. The lowest-energy manifold with $E_{i,\alpha=0}$, corresponds to those states in which each 1D system in the array is balanced between up- and down-spins and thus has $S_z = 0$, and $i$ indexes the states within this manifold. That is, in this manifold there are no broken pairs. The high-energy manifold $E_{i,\alpha=1}$ is at least $\Delta E_p$ above the low-energy manifold, corresponding to excited states with at least one broken pair, i.e., where the pair-constituents have moved onto separate chains. In the perturbative regime, we thus assume

$$|E_{i,\alpha} - E_{j,\alpha}| \ll |E_{i,\alpha} - E_{j,\beta}|; \quad \alpha \neq \beta \tag{7}$$

to hold.

We therefore target a small transverse hopping strength $t_\perp$ with respect to $\Delta E_s$ and $\Delta E_p$. Introducing the projector onto the lowest-energy manifold $\hat{P}_0 = \sum_i |E_{i,0}\rangle\langle E_{i,0}|$, the second-order perturbation theory for Hamiltonian Eq. (1) yields:

$$\hat{H}_{\text{eff}}^0 = \hat{P}_0 \hat{H}_0 \hat{P}_0 - \frac{t_\perp^2}{\Delta E_p} \hat{P}_0 \hat{H}_\perp^2 \hat{P}_0 . \tag{8}$$

Written explicitly, $\hat{H}_\perp^2$ is

$$\hat{H}_\perp^2 = \sum_{n,m=1}^{L} \sum_{\sigma \in \{\uparrow,\downarrow\}} \sum_{\{\mathbf{R}_i\}} \sum_{\hat{\mathbf{a}} \in \{\hat{\mathbf{y}},\hat{\mathbf{z}}\}} \left( \hat{c}_{n,\mathbf{R}_i+\hat{\mathbf{a}},\sigma}^\dagger \hat{c}_{n,\mathbf{R}_i,\sigma} \hat{c}_{m,\mathbf{R}_i+\hat{\mathbf{a}},-\sigma}^\dagger \hat{c}_{m,\mathbf{R}_i,-\sigma} + \text{h.c.}\right)$$

$$+ \sum_{n,m=1}^{L} \sum_{\sigma \in \{\uparrow,\downarrow\}} \sum_{\{\mathbf{R}_i\}} \sum_{\hat{\mathbf{a}} \in \{\hat{\mathbf{y}},\hat{\mathbf{z}}\}} \left( \hat{c}_{n,\mathbf{R}_i+\hat{\mathbf{a}},\sigma}^\dagger \hat{c}_{n,\mathbf{R}_i,\sigma} \hat{c}_{m,\mathbf{R}_i,\sigma}^\dagger \hat{c}_{m,\mathbf{R}_i+\hat{\mathbf{a}},\sigma} + \text{h.c.}\right) \tag{9}$$

$$= \hat{H}_{\text{pair}} + \hat{H}_{\text{exc}} . \tag{10}$$

Within Eq. (10), we identify two contributions, namely a pairing term $\hat{H}_{\text{pair}}$, which denotes the hopping of electron-electron pairs of opposite spin between neighboring chains and an exchange term $\hat{H}_{\text{exc}}$, denoting the exchange of particles of the same spin between neighboring chains.

In the following we use MF theory to eliminate the non-1D nature of $\hat{H}_\perp^2$. Here, we make use of the relation

$$c_i^{(\dagger)} c_j^{(\dagger)} = \left( c_i^{(\dagger)} c_j^{(\dagger)} - \langle c_i^{(\dagger)} c_j^{(\dagger)}\rangle \right) + \langle c_i^{(\dagger)} c_j^{(\dagger)}\rangle , \tag{11}$$

and assume $\left( c_i^{(\dagger)} c_j^{(\dagger)} - \langle c_i^{(\dagger)} c_j^{(\dagger)} \rangle \right)$ to be small. We, moreover, assume

$$\langle \hat{c}_{n,\uparrow} \hat{c}_{m,\downarrow} \rangle = \langle \hat{c}_{n,\mathbf{R}_i,\uparrow} \hat{c}_{m,\mathbf{R}_i,\downarrow} \rangle = \langle \hat{c}_{n,\mathbf{R}_i+\hat{\mathbf{a}},\uparrow} \hat{c}_{m,\mathbf{R}_i+\hat{\mathbf{a}},\downarrow} \rangle \ , \tag{12}$$

which means that all the chains are exact copies of each other. We end up with an effectively 1D expression for a Hamiltonian describing a higher-dimensional model, namely

$$\hat{H}_{\text{eff}}^{\text{MF}} = \hat{H}_0 - \sum_{n,m=1}^{L} \left( \alpha_{n,m}^* \hat{c}_{n,\uparrow} \hat{c}_{m,\downarrow} + \alpha_{n,m} \hat{c}_{m,\downarrow}^\dagger \hat{c}_{n,\uparrow}^\dagger \right)$$
$$+ \sum_{n=1}^{L} \sum_{\sigma \in \{\uparrow,\downarrow\}} \sum_{r=1}^{L-n} \left( \beta_{n,r,\sigma}^* \hat{c}_{n+r,\sigma}^\dagger \hat{c}_{n,\sigma} + \beta_{n,r,\sigma} \hat{c}_{n,\sigma}^\dagger \hat{c}_{n+r,\sigma} \right) \tag{13}$$

with

$$\alpha_{n,m} = \frac{2z_c t_\perp^2}{\Delta E_p} \langle \hat{c}_{n,\uparrow} \hat{c}_{m,\downarrow} \rangle \quad \text{and} \tag{14}$$

$$\beta_{n,r,\sigma} = \frac{2z_c t_\perp^2}{\Delta E_p} v \langle \hat{c}_{n+r,\sigma}^\dagger \hat{c}_{n,\sigma} \rangle \ , \tag{15}$$

and thus identify $\alpha_{n,m}$ with the MF-approximated pairing part of Eq. (10) and $\beta_{n,r,\sigma}$ with its exchange part. Here, we introduced the coordination number $z_c$, which denotes the number of neighboring chains. In our case $z_c = 4$, as the chains are assembled into a 2D square grid in the $\hat{\mathbf{y}} - \hat{\mathbf{z}}$-plane. The parameters $\alpha_{n,m}$ and $\beta_{n,r,\sigma}$ are the so-called MF parameters, meaning they need to be calculated self-consistently for all times. The work in [25] explains this for the ground state and for the finite-temperature equilibrium of the 3D system.

Since the present work aims to test and benchmark the real-time dynamical version of MPS+MF itself, in the following we are working with the simplest possible version of the Hamiltonian Eq. (13). We neglect the exchange term $\beta_{n,r,\sigma}$ and allow only for site-independent onsite pairing, meaning $\alpha_{n,m} \equiv \alpha_{n,n} \equiv \alpha$. This leads to

$$\hat{H}_{\text{eff}}^{\text{MF}} = \hat{H}_0 - \sum_n \left( \alpha^* \hat{c}_{n,\uparrow} \hat{c}_{n,\downarrow} + \alpha \hat{c}_{n,\downarrow}^\dagger \hat{c}_{n,\uparrow}^\dagger \right) \tag{16}$$

with

$$\alpha = \frac{1}{L} \frac{2z_c t_\perp^2}{\Delta E_p} \sum_{n=1}^{L} \langle \hat{c}_{n,\uparrow} \hat{c}_{n,\downarrow} \rangle \ . \tag{17}$$

In this last expression we are adapting the evaluation of the order parameter $\alpha$ to the open boundary conditions. Obtaining $\alpha$ from an average across the entire system removes the spatial variation that is solely due to these open boundaries.

Regarding the reliability of the partial MF decoupling in the two perpendicular weak-tunneling directions, we expect that properties like the order parameter will inevitably be overestimated, as in any MF theory. For the equilibrium case, several of the authors demonstrated that the MPS+MF approach produces the correct physics compared against QMC, in regimes in which the latter approach is quasi-exact, in a negative-$U$ Hubbard model on a 2D square lattice with anisotropic tunneling [25]. That work also shows that the error in $T_c$ for the SC within the MPS+MF framework is a quasi-constant one in $t_\perp$ over a significant range. Moreover, at zero temperature, the overestimation of the SC order parameter becomes systematically better as $t_\perp$ decreases. We also point out that

the degree of overestimation decreases strongly as the dimensionality of the system grows, as expected from the concurrent decrease of quantum and thermal fluctuations, which our MF-treatment partially neglects. In equilibrium, we know this due to work by some of the present authors [26], applying the MPS+MF framework to 3D lattice-bosons, which yields much more modest overestimations of key quantities such as $T_c$ for superfluidity compared to the 2D case [2]. Naturally, the good performance of the MPS+MF framework in equilibrium systems and especially in 3D by itself does not guarantee comparable performance to the non-equilibrium systems studied here. The question is thus how the framework could be validated independently. Unfortunately, while the performance for equilibrium systems could be checked in-depth for bosons, and in limiting cases for fermions, using QMC-techniques, there are no other classical computational methods that can currently reach the system sizes and time scales of the framework shown in this work. But this would be essential for any meaningful comparison to our approach, as is clear from the data shown in Sec. 4. There, it is plain that dynamically induced SC from a non-SC starting state will only build up over time scales that are well over an order of magnitude larger than $t^{-1}$, and that for 3D systems could only set in at $\mathcal{O}(10^2)$ sites, given the SC healing length will be on the scale of several sites at least in $\hat{\mathbf{x}}$-direction, and well larger than that in the two other directions. If the linear dimension of the cluster dropped below these scales, the small size could easily preclude any SC. As discussed in Sec. 1, the existing alternatives could not reach such time and length scales due either to the growth of entanglement entropy (for techniques based solely on MPS), or the fermionic sign problem (for QMC-based ones), even for special or limiting cases. Fortunately, there is a powerful alternative for checking the performance of the scheme free from these constraints, discussed in detail in Sec. 5, which applies to that subclass of models limited to purely on-site interactions. For these, existing experiments on ultracold atomic lattice gases already offer all the features that would be required to independently verify the predictions of the dynamical MPS+MF framework. For the scope of the present work, that leaves internal consistency checks, which are done as part of the case study in Sec. 4.

## 3 MPS+MF-Algorithm for self-consistent time evolution

The expectation values needed to compute the MF parameter $\alpha$ in Eq. (17) are calculated using a self-consistent scheme for both the time evolution and for the ground-state search of our model system. In this section a schematic description of the time-evolution routine is presented, which is one of our main results. The algorithm is based on the work of H. Strand et al. published in [34], where a non-equilibrium version of real-time DMFT for bosons is introduced. Our work incorporates this real-time scheme into a MPS framework and adapts it to 3D lattices of correlated fermions built from weakly coupled 1D systems. All results obtained in the following were generated with Ian McCulloch's matrix product toolkit [35] using its time-evolving block decimation (TEBD) implementation. Note that the described algorithm is not limited to TEBD, but also other MPS based time-evolution methods [36] can be used instead. The initial ground states from which the time evolution proceeds were generated from a self-consistent scheme introduced by Bollmark et al. in [26], which is also briefly described in App. A.

At the beginning of each time step, we start with a state $|\psi(t_1)\rangle$ at time $t_1$, which we already have obtained before (either as a previous step or as initial state). From this state, we measure the value of the MF parameter $\alpha(t_1)$. Now, we guess which value $\alpha$ might take

---

[2]While the 2D case concerned lattice fermions, these had strong onsite attraction, and would thus be close to effective hard-core bosons with residual nearest-neighbour repulsion.

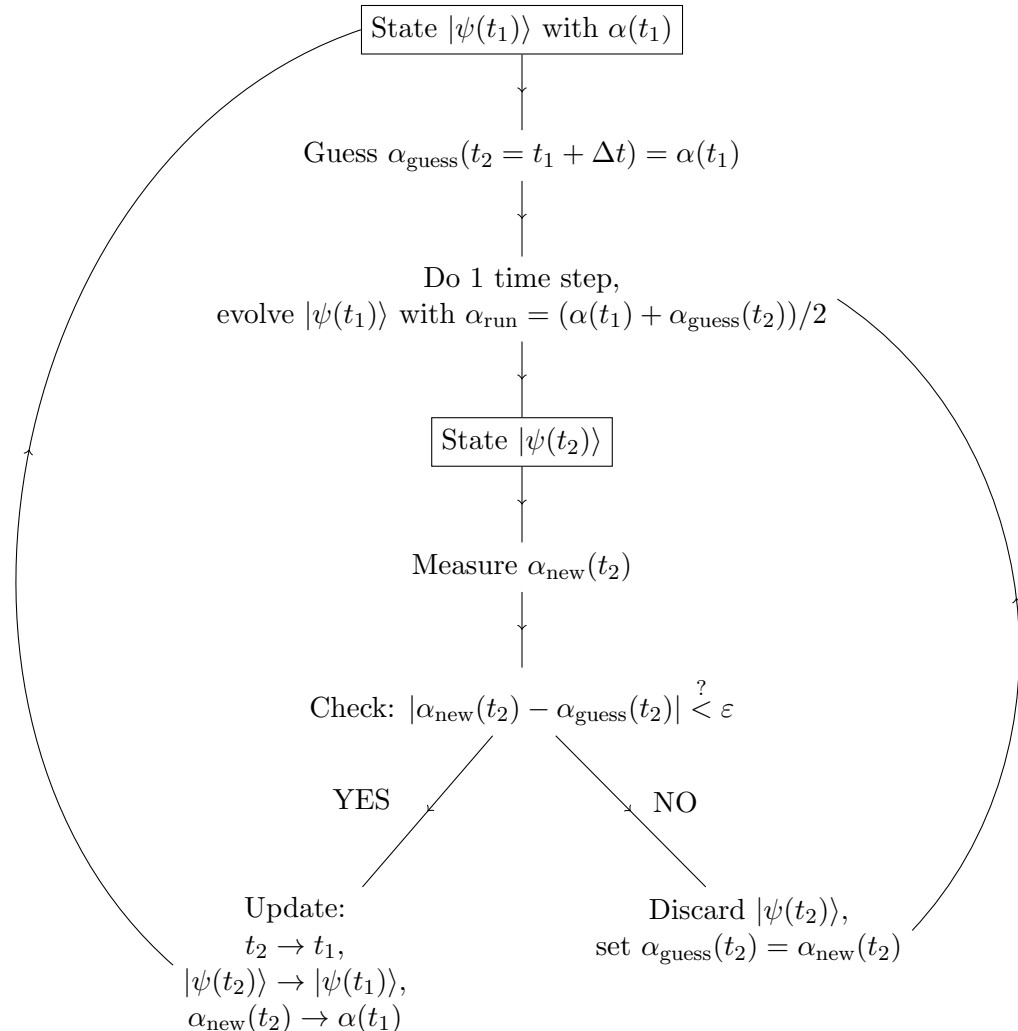

Figure 2: Self consistency loop for one time step. As the MF-parameter $\alpha(t)$ depends on the state $|\psi(t)\rangle$ itself, a continuous yet self-consistent adjustment of it is required. Our scheme achieves this at each discrete time step the algorithm advances, by trying to evolve with an equally weighted average of the current value of $\alpha(t)$ and an updated $\alpha$-value, which, in the first attempt is just a heuristic guess. If the measured $\alpha$-value of the new state thus evolved does not match the updated $\alpha$-value (very likely in the early loops) up to some pre-defined precision $\varepsilon$, the wavefunction is discarded, and a new attempt is made, this time with the just-measured $\alpha$-value as the new guess for the updated $\alpha$.

after one discrete time step $\mathrm{d}t$. In this work, at the start of the self-consistency iterations for each time step, we just assume that the $\alpha$ value does not change at all. In any case, the guess for $\alpha$ at $t_2 = t_1 + \mathrm{d}t$, is labeled $\alpha_{\mathrm{guess}}(t_2)$. Then, we evolve the system from $t_1$ to $t_2$ using the mean of $\alpha(t_1)$ and $\alpha_{\mathrm{guess}}$. From the resulting tentative $|\psi(t_2)\rangle$ we can once again measure the MF parameter $\alpha_{\mathrm{new}}(t_2)$. Next we calculate the distance between the measured and the guessed value and compare it to a chosen precision $\varepsilon$,

$$|\alpha_{\mathrm{new}}(t_2) - \alpha_{\mathrm{guess}}(t_2)| < \varepsilon \quad \text{with} \quad \varepsilon \ll 1 . \tag{18}$$

For all data shown in this work, we have set $\varepsilon = 10^{-12}$. If Eq. (18) is fulfilled, we keep the state $|\psi(t_2)\rangle$ and proceed with the next time step. Otherwise, we discard $|\psi(t_2)\rangle$ and repeat the time step using the mean of $\alpha(t_1)$ and $\alpha_{\mathrm{new}}(t_2)$. The loop is repeated until

Eq. (18) is fulfilled. A schematic of the algorithm is depicted in Fig. 2.

# 4 Transient superconductivity after a fast ramp of the nearest-neighbor interaction

In this section, we present our results using the self-consistent MPS+MF scheme and find that in the extended Hubbard model Eq. (1) the BCS order parameter for SC grows in time and begins to oscillate around a finite value on the treated time scales. This indicates the formation of transient SC, which is the second main result of this paper. In the following, all parameters are measured in units of the hopping parameter $t \equiv 1$.

More specifically, we follow Paeckel et al. [19] and tune the system's parameters from an insulating charge-density wave (CDW) phase into a SC phase. However, we find that the sudden quench performed in [19] is numerically more challenging[3] within the self-consistent scheme (see App. B), so we instead perform a fast ramp.

In order to check the equilibrium phases of the 3D model we use the self-consistent MPS+MF approach to compute the ground states using the routine introduced by Boll-mark et al. [26] for different parameters and measure the expectation value of the MF parameter $\alpha$. We find that for $t_\perp = 0.2$, $U = -4$ and $V = 0.25$ the system possesses the main properties of a CDW phase relevant for us, i.e., we find alternating occupation of the lattice sites by the electrons and a vanishing value of $\alpha$. For $U = -4$ and $V = -0.25$ instead, the system is SC, as here $\alpha \sim 10^{-1}$ becomes finite and density oscillations less pronounced. These are the same parameters treated by Paeckel et al. in [19] for the purely 1D system. Hence, we perform a fast ramp by tuning the values of the nearest-neighbor interaction from $V = 0.25$ to $V = -0.25$ as further detailed below.

Since the effective Hamiltonian Eq. (16) depends on the MF parameter $\alpha(t)$ the question of how to choose $\alpha_{\mathrm{ini}} := \alpha(t = 0)$ arises. For the CDW system $\alpha = 0$ and it is hence difficult for it to grow with the method outlined in Fig. 2. Because of this, unless otherwise noted, our default value for this work is $\alpha_{\mathrm{ini}} = 10^{-4}/\mathrm{d}t$, where $\mathrm{d}t$ is the size of the discretized time step of the simulation. Such a small yet finite value is justified by the fact that any system will either have a microscopic fraction of pairs in the center-of-mass zero-momentum state to begin with, or such a fraction is generated during the ramp or quench. Scaling $\alpha_{\mathrm{ini}}$ inversely in $\mathrm{d}t$ ensures that simulations with different $\mathrm{d}t$ agree over long times, see Fig. 3.

The MF term of the Hamiltonian causes the effective model to be no longer particle-number conserving, hence, we need to adjust the value of the chemical potential $\mu$ corresponding to the system size and to the onsite repulsion $U$ in order to fix the average density of the total system. From the ground-state calculations we find the values of $\mu$ that are listed in table 1. We keep the values of $\mu$, determined in this manner, fixed throughout the whole time evolution in order to keep our algorithm simple and stable. However, we still need to keep track of the overall density of our system during the time evolution to check if this assumption of a time-independent chemical potential is justified. Indeed, for our simulations, the value of the density is preserved to a good accuracy over the time scales treated by us (see Figs. 3 and 4). In general, however, it might be necessary to also include a variation of $\mu$ into the self-consistency scheme.

---

[3]Instant quenches are amongst the most demanding use-cases for any real-time MPS-based simulations.

Table 1: List of values for the chemical potential $\mu$ to obtain half filling for $U = -4.0$ and $V = \pm 0.25$ for various system sizes $L$.

| $L$ | 12 | 20 | 30 |
|---|---|---|---|
| $\mu(V = -0.25)$ | -2.44 | -2.47 | -2.48 |
| $\mu(V = 0.25)$ | -1.66 | -1.63 | -1.62 |

## 4.1 Time evolution of the BCS order parameter and of the total energy

In the following, we investigate the time evolution of the BCS order parameter $\alpha(t)$ (see Eq. (17)) and of the total energy $E(t)$ of the system. The latter cannot be expected to remain constant as the MF term changes the Hamiltonian Eq. (16) during evolution. In addition, we monitor the total density of the system, which should stay at a value of $\rho = 1$ (half filling) during the whole time evolution.

Since we find fast ramps to have lower errors over the simulated time windows than instantaneous quenches, we linearly decrease the value of the nearest-neighbor interaction $V$ from $V = 0.25$ to $V = -0.25$ within a time window of $\Delta t_{\mathrm{ramp}} = 3.0$. A more detailed discussion of the effect of the size of the time window $\Delta t_{\mathrm{ramp}}$ can be found in App. B. In Fig. 3 we see the results for a 30-site system for an evolution up to time $t_{\mathrm{end}} = 50$. Since $\alpha(t)$ is complex-valued we show the evolution of the magnitude $|\alpha(t)|$ and of the phase $\varphi(t)$ of the order parameter in Figs. 3 to 5. We find that $|\alpha(t)|$ grows up to time $t \sim 45$ to a value of approximately $|\alpha| \approx 0.06$, which is clearly non vanishing and hence indicates the formation of a non-equilibrium SC state. In contrast, if we consider a time evolution without a quench or ramp, i.e., $V = 0.25$ during the whole evolution, the value of $\alpha$ stays unchanged at an order of magnitude of $10^{-5}$ throughout the whole time evolution as can be seen by the dotted blue lines in Fig. 3. The phase $\varphi(t)$ decreases as long as $V$ is decreasing, then oscillates around a value of approximately $\varphi(\alpha)/\pi \approx -0.8$ and seems to increase again slightly when $|\alpha|$ has reached its maximum. We interpret this behavior as an expression of a Josephson effect in-between 1D chains to the extent it can be captured by a single 1D system with time-evolving MF amplitudes. As a kernel of SC order manifests itself in the different chains of the 2D array the macroscopic phases of SC states, within each chain, will be initially uncorrelated, then start aligning via the Josephson effect. With density fluctuating within each individual chain the Josephson effect will keep the phase fluctuating while the system finds a new equilibrium after the rapid ramp, as Fig. 3b shows.

In Fig. 3d we show the evolution of the total energy per site $E(t)/L$ and in Fig. 3e the deviation of the total density $\rho(t)$ from the desired value $\rho_{\mathrm{target}} = 1$. We find that this deviation is of the order of $3 \cdot 10^{-5}$ or smaller for all the times treated, indicating that keeping the chemical potential $\mu$ fixed leads only to small errors. The total energy per site $E/L$ behaves as expected during the ramp and decreases almost linearly for the duration of the ramp. Afterwards, we first observe a nearly constant behavior, then a strong decrease until a minimum at time $t \approx 45$, shown in the inset of Fig. 3a. We read the behavior of $E(t)/L$, especially at long times, as the system starting to further lower its energy through condensing Cooper pairs, as the drop in $E(t)/L$ coincides markedly with the onset of a finite value of $\alpha(t)$. We also study the effect of system size, to make certain the dynamical onset of SC would survive in the thermodynamic limit. In Fig. 4 we compare the results for different chain lengths $L$. From these, we extract the instant $t_{\mathrm{SC}}$, at which $|\alpha(t)|$ reaches its first maximum. The data of the 12-site system shows the onset of oscillation for $|\alpha(t)|$ around a finite value, indicating a dynamically induced SC phase (longer-time simulations for $L = 12$ further confirm this, as shown in Figs. 5 to 7 for times

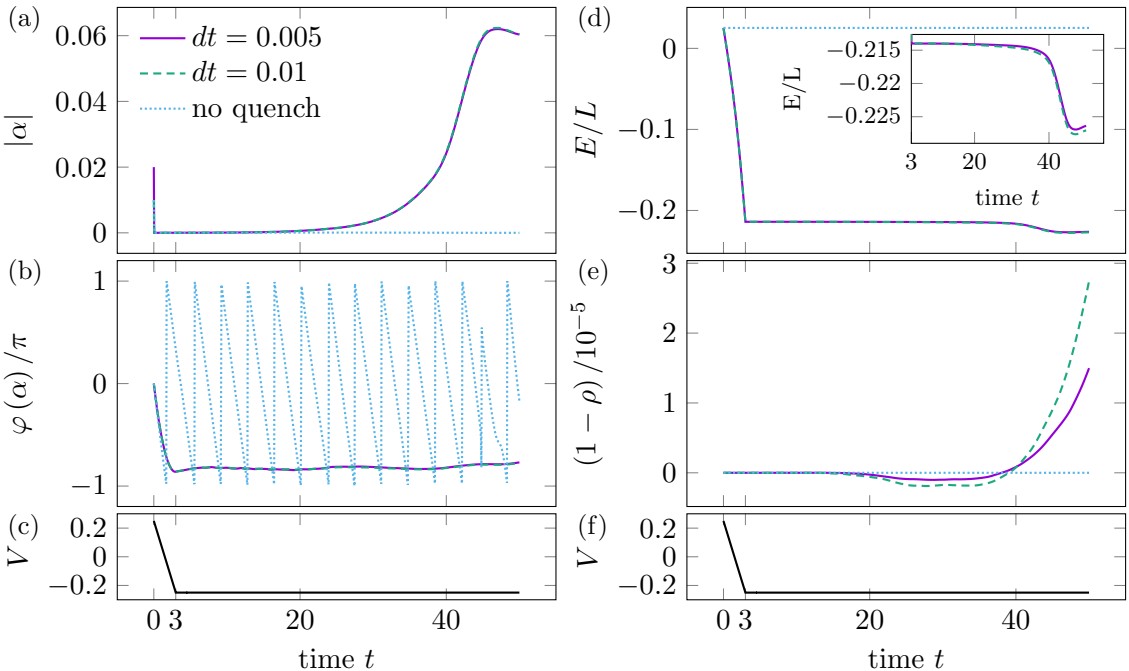

Figure 3: Evolution of the considered parameters in time during and after a ramp on a 30-site system. The plots at the bottom ((c) and (f)) show the nearest-neighbor interaction, which decreases from $V = 0.25$ to $V = -0.25$ during a time window of $\Delta t_{\text{ramp}} = 3.0$. Graphs (a) and (b) show the evolution of the MF parameter $\alpha$ split up into magnitude and phase. Graphs (d) and (e) show the evolution of the total energy per site of the system and of the total density. The inset in (d) shows the evolution of the energy per site after $V$ was decreased. The dip towards the end of the evolution coincides with the rapid increase in $|\alpha(t)|$, and signals the system overall lowering it's energy by entering a SC state. The legend is valid for all plots. All the data shown here were obtained with a bond dimension of $\chi = 250$, an initial guess of the MF parameter of $\alpha_{\text{ini}} = 10^{-4}/\text{d}t$, and the chemical potential was taken from table 1. We compare the ramp scenario (solid violet and dashed green) with an evolution during which we keep the nearest neighbor interaction at $V = 0.25$ constant (dotted blue). Even though this state is given the same initial $\alpha$-value as a seed as in the ramp scenario, $\alpha(t)$ rapidly decays and remaining at near-constant and near-zero values. This shows that no SC state develops without a quench to parameters corresponding to SC in equilibrium. For this calculation we chose a time step of $\text{d}t = 0.01$.

up to $t_{\text{max}} = 100$). The inset of Fig. 5a displays an extrapolation in inverse chain length $1/L$ of $t_{\text{SC}}$. In order to see whether $t_{\text{SC}}$ diverges we performed a quadratic and a linear fit, both indicating a finite value in the limit $L \to \infty$. Since for the larger system sizes $|\alpha(t)|$ starts to oscillate at around the maximal time reached by us, it is difficult to obtain a finite-size extrapolation of the value of the SC order parameter. In order to do so, one needs to extend the simulations for the larger systems to substantially longer times, which is beyond the scope of this paper.

## 4.2 Accuracy of results and sensitivity to simulation parameters

The results so far were all obtained using the same parameters for the self-consistency cycle. In the following we study how sensitive the results are on parameters like the initial guess of the MF parameter $\alpha_{\text{ini}}$ (see Sec. 3), the bond dimension of the MPS calculations, or the discrete time step $\text{d}t$. To study these effects, we focus on the 12-site system in order

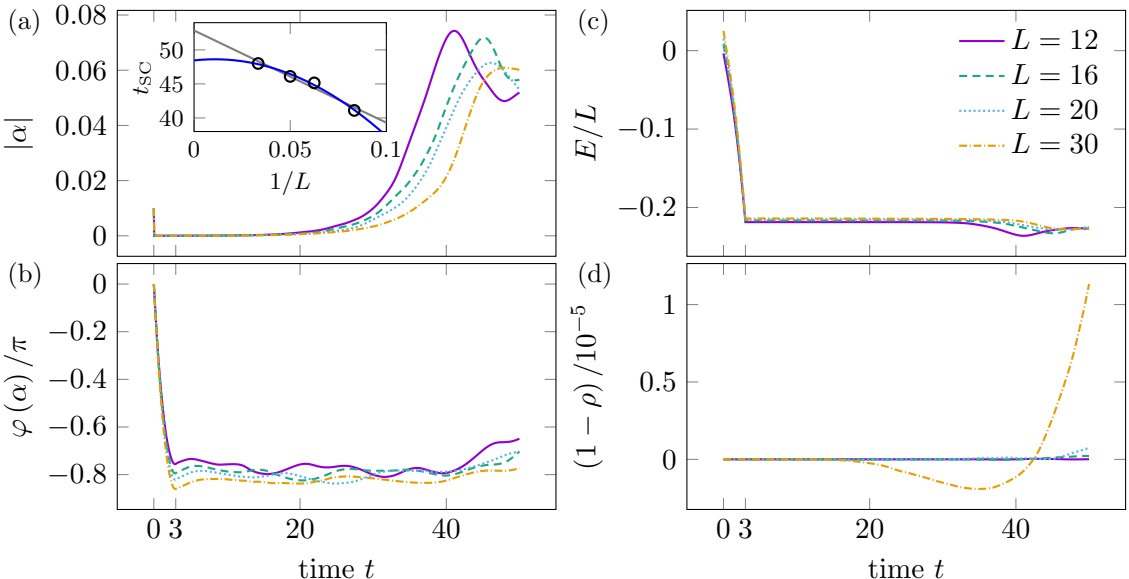

Figure 4: Evolution of SC order-parameter $\alpha$, energy per site $E/L$, and density $\rho$ in time during and after a ramp for sizes $L = 12, 16, 20, 30$. Magnitude and phase of $\alpha$ are shown in (a) and (b), respectively. The inset in (a) shows the time at which the first local maximum in $|\alpha|$ occurs plotted against the inverse of the system size $1/L$. Both the linear and the quadratic fit suggest a finite and comparable value of $t_{SC}$ in the limit $L \to \infty$. Evolution of the total energy of the system and the total density are shown in (c) and (d). All data were obtained with a bond dimension of $\chi = 500$, an initial guess of the MF parameter of $\alpha_{ini} = 10^{-4}/dt$, a ramp time window $\Delta t_{ramp} = 3$, and the chemical potential was taken from table 1.

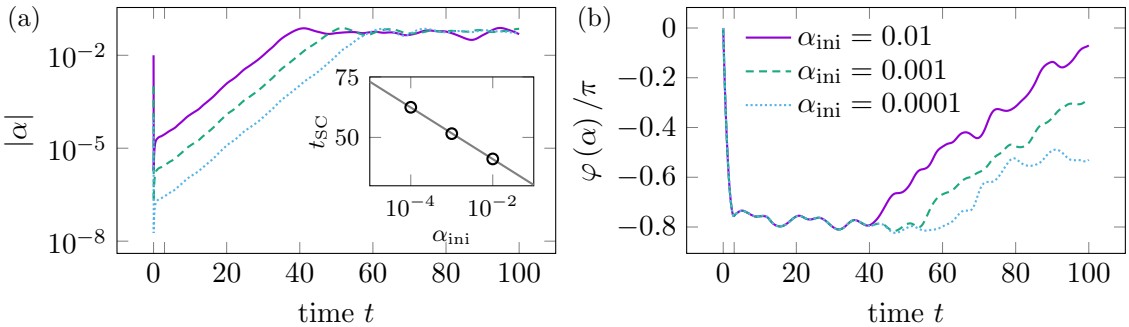

Figure 5: Evolution of the MF parameter $\alpha$ split up into its magnitude (a) and its phase (b) for different initial guesses $\alpha_{ini}$ in a 12-site system. We see that the reduction of $\alpha_{ini}$ induces merely a shift in the data, at least up to time $t_{SC}$ at which the first maximum of $|\alpha|$ occurs. The inset in (a) shows $t_{SC}$ vs. $\alpha_{ini}$ and a linear fit on a semilogarithmic scale. This shows that $t_{SC}$ grows merely logarithmically with $\alpha_{ini}$. The data shown was obtained with $\chi = 500$ and $dt = 0.01$, a ramp time window $\Delta t_{ramp} = 3$, and $\mu$ was taken from table 1. We stress that the evolution of $|\alpha|$ up to the first maximum at $t_{SC}$ is nearly identical for different $\alpha_{ini}$ up to a shift, as highlighted by the logarithmic scale. Beyond $t_{SC}$, curves for different $\alpha_{ini}$ oscillate around closely similar averages, and any deviations of the $\alpha(t)$-curves from each other are due to the inevitable finite-size effects, which will naturally depend on the values of $\alpha_{ini}$.

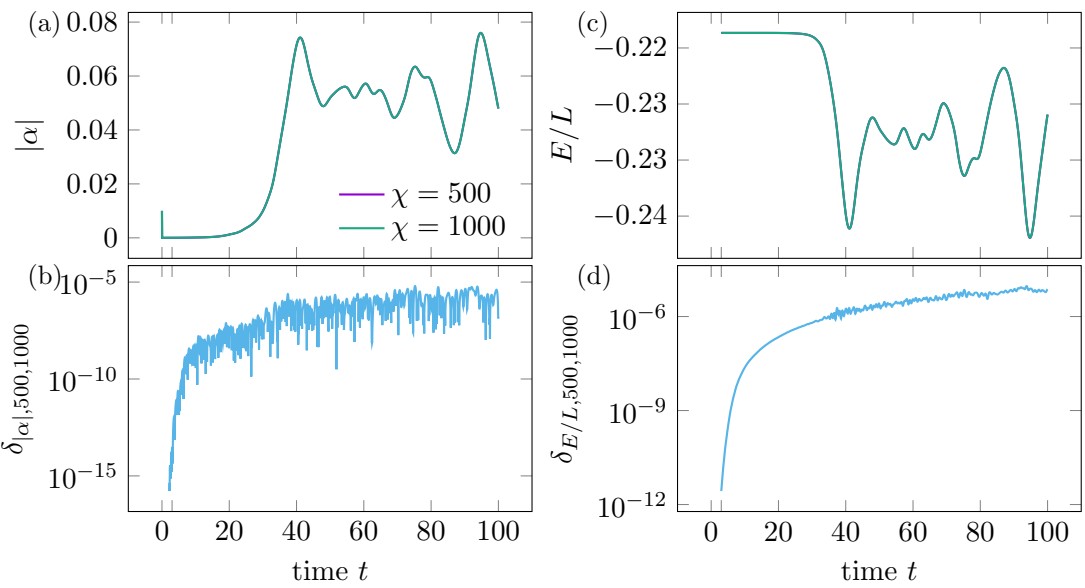

Figure 6: Evolution of (a) the magnitude of the MF parameter $|\alpha|$ and (c) the total energy $E$ for two different bond dimensions $\chi_1 = 500$ and $\chi_2 = 1000$ in a 12-site system. (b) and (d) show the difference $\delta_{\mathcal{O},\chi_1,\chi_2}$ between the observables we measure for these two different bond dimensions. All calculations were done with $\alpha_{\mathrm{ini}} = 10^{-4}/\mathrm{d}t$, a ramp time window $\Delta t_{\mathrm{ramp}} = 3$, and $\mathrm{d}t = 0.01$.

to reach the longest time scales.

Figure 5 shows the evolution of the magnitude and phase of $\alpha(t)$ for different initial values $\alpha_{\mathrm{ini}}$. Decreasing the value of $\alpha_{\mathrm{ini}}$ induces a shift of $t_{\mathrm{SC}}$ to later times. In order to further analyze this, we plot the value of $t_{\mathrm{SC}}$ against the value of $\alpha_{\mathrm{ini}}$ in the inset of Fig. 5a. Speaking to the soundness of our MF approximation, we find that $t_{\mathrm{SC}}$ increases only very weakly with $\alpha_{\mathrm{ini}}$, i.e., logarithmically. While this indicates a diverging time for the onset of SC order in the limit $\alpha_{\mathrm{ini}} \to 0$, this is merely consistent with $\alpha_{\mathrm{ini}} = 0$ being an unstable fix point of the dynamic MF algorithm in the regime we ramp into. But any finite value, even a microscopic one, will yield dynamically induced SC order in finite time when ramping into the SC parameter regime. And, as argued at the outset of Sec. 4, on general physical grounds there will always be some electron pairs whose center-of-mass momentum is zero.

In Fig. 3 we compare two different discretized time steps, $\mathrm{d}t = 0.005$ and $\mathrm{d}t = 0.01$, respectively. The results are nearly identical, only a small deviation of the total density, which agrees up to $\sim 10^{-5}$, can be seen in Fig. 3e.

Regarding the discarded weight of our simulations, we find that even for the smallest bond dimension these values stay below $10^{-6}$ within the time domains we consider. Nevertheless, we examine the dependence of our results on the MPS bond dimension $\chi$. For this purpose we compute the deviation of the value of an observable $\mathcal{O}$ for two different values of $\chi$,

$$\delta_{\mathcal{O},\chi_1,\chi_2} = |\mathcal{O}(\chi_1) - \mathcal{O}(\chi_2)| \ . \tag{19}$$

At any fixed value of $\alpha_{\mathrm{ini}}$ and $\mathrm{d}t$ we find this to be the most reliable estimator for the accuracy of our combined MPS+MF approach (assuming the latter parameter is chosen to be sufficiently small) and focus in the following on this quantity.

In Figs. 6 and 7 we present results for the observables $|\alpha(t)|$ and $E(t)$ obtained with two different bond dimensions $\chi_1 = 500$ and $\chi_2 = 1000$ for the 12-site system, and for $\chi = 250$

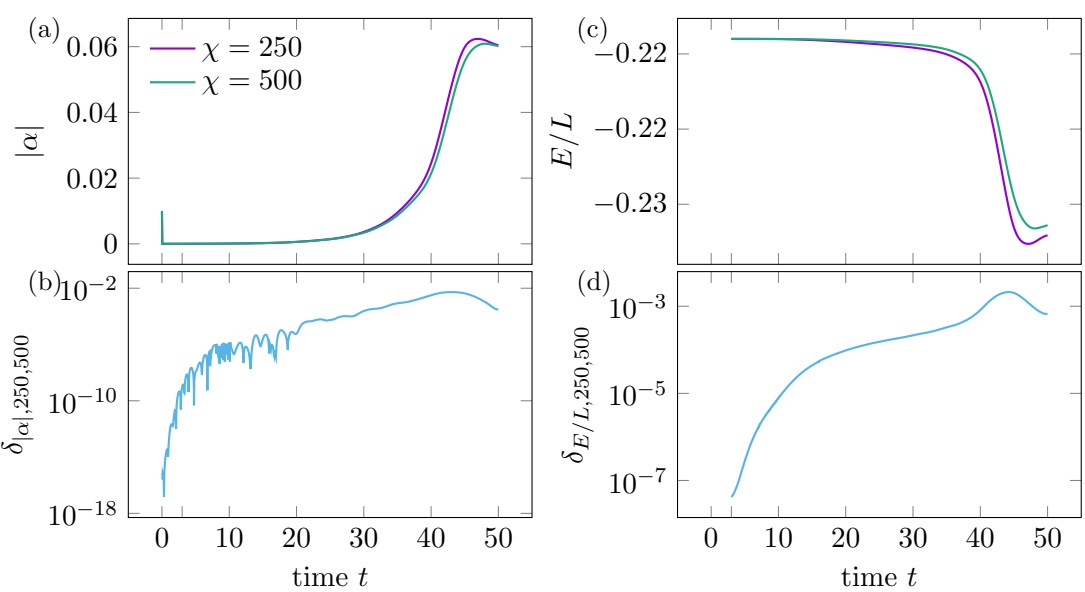

Figure 7: Analog to Fig. 6 but for a 30-site system and for bond dimensions $\chi_1 = 250$ and $\chi_2 = 500$.

and $\chi = 500$ for the 30-site system, respectively, and also the difference of the respective results. For the larger system it was necessary to substantially reduce the values of $\chi$, since otherwise the numerical expenses would exceed the available resources. We find that the deviation of the results is $\sim 10^{-6}$ for the values of $|\alpha(t)|$ and $\sim 10^{-4}$ for the total energy $E(t)$, in the case of the 12-site system. For both observables, this is small compared to the order of magnitude of the observables themselves, so that we conclude these values of $\chi$ suffice to provide quantitatively accurately results, within the dynamical MPS+MF framework.

For the 30-site system, however, the deviation is $\sim 10^{-3}$ for $|\alpha(t)|$ and $\sim 10^{-2}$ for $E(t)$. This is rather large in comparison to the order of magnitude of the observables themselves. The data obtained from these calculations is hence only trustworthy in regards to the qualitative physics, but for the larger chain lengths one needs a larger bond dimension to obtain a better quantitative convergence of the results.

## 5   Conclusion

This work presents a self-consistent real-time MPS+MF approach for investigating the time evolution of a 3D extended Hubbard model after a fast ramp. By combining perturbation theory with a MF ansatz, we construct an effective 1D Hamiltonian Eq. (13) capable of capturing the dynamical build-up of SC correlations for this 3D model system, when quenching or rapidly ramping into a Hamiltonian parameter regime corresponding to SC order in equilibrium. This approach is generic to any 3D system composed out of gapped 1D systems of fermions, as long as coupling between 1D systems is sufficiently weak for single-fermion tunneling in-between 1D systems to be suppressed. For concrete demonstration of the performance of this approach, we chose systems of 1D extended Hubbard chains, arranged in parallel in a 2D square array, forming a 3D system with weak interchain tunneling $t_\perp$, negative onsite repulsion $U$, and nearest-neighbor interaction $V$ along each chain.

We benchmark the self-consistent algorithm introduced on the simplest possible version

Eq. (16) of the resulting effective MF Hamiltonian, only taking onsite pairing into account and neglecting the particle-hole terms Eq. (15). We test our approach on systems where each chain is up to $L = 30$ sites long. Using this algorithm we compute the time evolution of the BCS order parameter for SC order $\alpha(t)$, as a direct indicator of dynamically induced SC. The results show that SC order sets in after a fast ramp from $V = 0.25$ to $V = -0.25$, where the initial $V$-value realizes an insulating CDW state, and the final value would correspond to SC order at equilibrium. These results are broadly comparable to previous 1D results [19] and represent a best-case scenario, in which double occupancies already present in the CDW help to form the non-equilibrium SC state after the ramp.

Performing infinite-size extrapolations and studying the effect of the microscopic initial kernel of SC order $\alpha_{\mathrm{ini}}$ shows that dynamically induced SC is not merely a trivial size effect, but actually present in the thermodynamic limit, and even the smallest yet finite magnitude for $\alpha_{\mathrm{ini}}$ will result in establishing order within a finite window of time. At the same time, we find that resource requirements increase substantially with chain length $L$, but several tens of sites and time frames between one and two orders of magnitude in units of inverse fermion tunneling $t^{-1}$ are accessible already with the modest resources employed for the present proof-of-principle work.

The present work presents multiple avenues for interesting and potentially valuable follow-up work. One of these would be to move towards a regime that is physically more realistic as far as solid state systems are concerned, in which the pair-binding energies $\Delta E_p$ would be significantly smaller than in the present work. This would entail either lowering $U$, or working directly with a 1D model offering repulsively mediated pairing, such as a doped two-leg Hubbard ladder [37,38]. This would require retaining more particle-particle terms Eq. (14) than we have done for the present proof-of-principle, as well as incorporating the particle-hole terms Eq. (15) into the self-consistent time-evolution step, see Fig. 2. This would be straightforward, as a generic ansatz for the first iteration of these terms is practically imposed by the physics of these 1D systems. As detailed in, e.g., [39], both classes of terms decay with an exponential envelope function characterized by the spin-correlation length, which in turn is easy to obtain from static correlators via density-matrix renormalization group (DMRG) [40,41] calculations for the isolated systems. Finally, we observe that while in the present work the initial state was a pure zero-temperature one, extensions of our new approach to finite temperature would be entirely straightforward. Being ultimately MPS-based, it can tap into practically all previous refinements of MPS-numerics, such as the simulation of non-equilibrium dynamics of finite-temperature states that minimizes the growth of entanglement with time [42].

With the above refinements, our approach opens new and potentially exciting avenues for advancing the understanding of dynamically induced SC in solids jointly with the domain of analog quantum simulation. Existing experiments with ultracold atomic lattice gases of fermions in 3D already offer the capability of realizing arrays of weakly coupled negative-$U$ Hubbard chains initially in the analogue of the PG regime. Results from ref. [25] (c.f. Fig. 11a there) illustrate that with today's experimental control over the parameters of these lattice-gas experiments such a regime can practically always be found. That is, inside each chain up- and down-pseudospin atoms would be paired, with an associated pairing energy $\Delta E_p$, from the effective attractive on-site interaction, but the coupling $t_\perp$ between these chains would be deliberately chosen to be too weak for the 3D array to enter the SC regime at the working temperature of the experiment. In this manner, these analog quantum simulators would mimic the basic presumed initial conditions of the layered high-$T_c$ compounds in pump-probe experiments on dynamically induced SC. However, there would be two key differences to the solid state: the active sub-units would be 1D chains instead of 2D planes, and as a result of that these analog quantum simulations

would be amenable to be modeled by quantitative many body numerics, i.e., the approach developed in the present work and incorporating some of the refinements laid out in the preceding paragraphs, in a way that the solid state experiments are not. Then $t_\perp$ would be quenched or rapidly increased to a value that at equilibrium would correspond to the SC regime, analogous to other types of experimentally performed rapid parameter changes (c.f. [43,44] and references therein). Any SC emerging dynamically from this change in $t_\perp$ could be directly tracked in real time and unambiguously via the coherence peak of paired atoms, which would constitute direct proof of the onset of SC off-diagonal long-range order in these 3D ultracold atomic lattice gases [45]. The dynamical MPS+MF framework developed here could then be used to directly model these experiments, ascertaining whether any predicted onset of dynamically induced SC qualitatively and quantitatively matches the observations. It would further allow to check whether the natural choice for initializing $\alpha_{\mathrm{ini}}$, the residual pairing correlations measured experimentally for the initial non-SC state, would yield the correct time scale for SC to emerge dynamically. This modeling would thus serve the dual purpose of validating the numerical framework itself, which would be necessary as there is currently no alternative classical numerical approach that could approach the system sizes, time scales and temperatures accessible to our dynamical MPS+MF approach. In this way, comparison between experiment and our theory framework would also advance the field of analog quantum simulation of demanding non-equilibrium many-body problems. Once these comparisons work sufficiently well, a natural extension would be to advance to experiments in which 2D planes of negative-$U$ Hubbard models are initially weakly coupled with a $t_\perp$-term, s.t. one would again start in a PG-regime, but this time even closer to the basic physics at the root of the experiments of the layered high-$T_c$ materials. Anchored by the previous experiments on the quasi 1D arrays and the quantitative comparison made possible by the theory framework developed here, the outcomes of such a second stage of experiments could provide a strong input to the ongoing discussion as to how to interpret experiments on dynamically ind¸uced SC in layered high-$T_c$ materials.

**Acknowledgements**   We acknowledge helpful discussions with Hugo Strand, Sebastian Paeckel, and Oscar Grånäs. A.K. would like to thank Michael Fechner and Michele Buzzi from the group of Andrea Cavalleri at MPSD Hamburg for helpful discussions and their hospitality. We acknowledges financial support by the ERC Starting Grant from the European Union's Horizon 2020 research and innovation program under grant agreement No. 758935. SRM and SM acknowledge funding by the Deutsche Forschungsgemeinschaft (DFG, German Research Foundation) - 217133147/SFB 1073, project B03 and TU Clausthal. We also acknowledge access to computational resources provided by the GWDG. This work also used the Cirrus UK National Tier-2 HPC Service at EPCC (http://www.cirrus.ac.uk) funded by the University of Edinburgh and EPSRC (EP/P020267/1).

## A   Self Consistent Ground State Search

As mentioned in Sec. 4 of this paper, we are making use of the self-consistent ground-state search developed by Bollmark et al. [25, 26]. Here a brief description of this algorithm shall be given.

Basically, a ground-state search in MPS language is an optimization problem solved via DMRG. However, in our case we are dealing with the special case that not only the state $|\psi\rangle$ has to be optimized but that we also do not know all parameters of the Hamiltonian as one of the parameters, namely $\alpha$, depends on the ground state itself. This

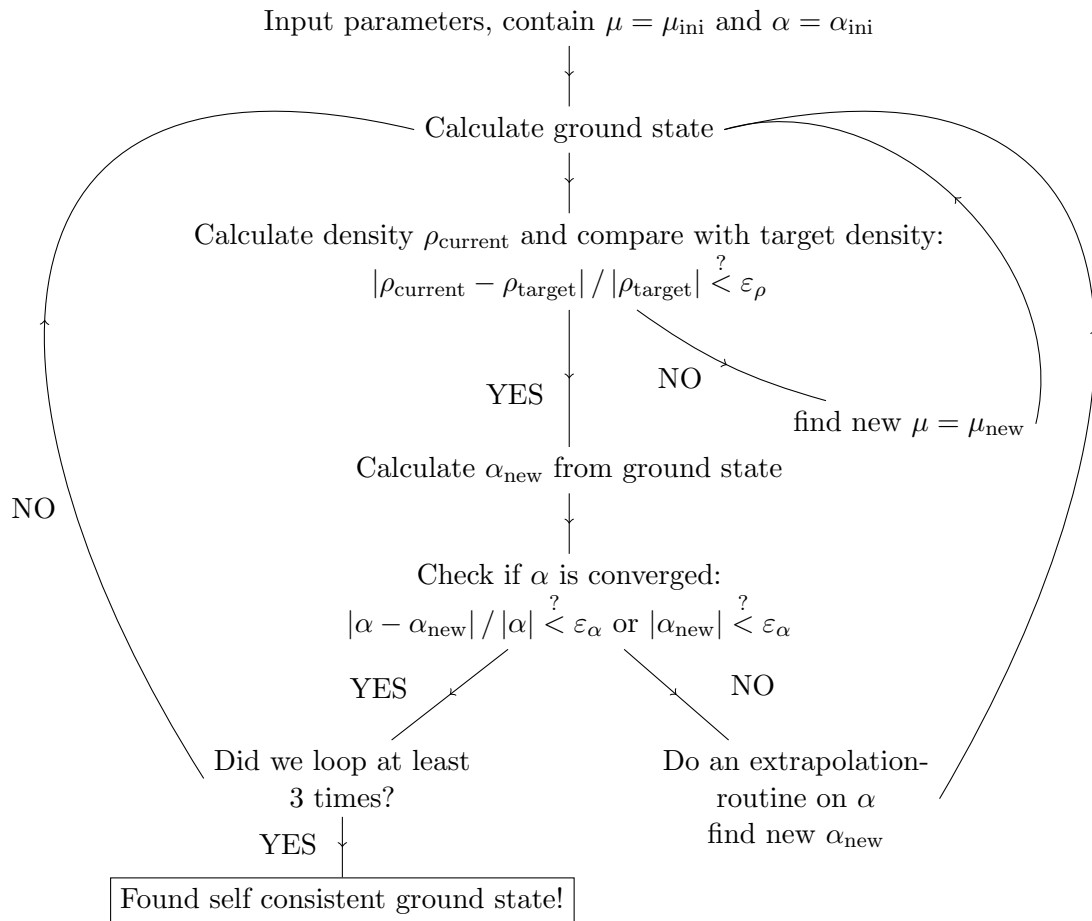

Figure 8: Self consistency loop for the ground-state search. As the MF-parameter $\alpha$ depends on the ground state itself, it has to be adjusted after each DMRG step. As the effective Hamiltonian, furthermore, is no longer particle number conserving we also need to update the chemical potential $\mu$ continuously.

is why we need to adjust this parameter iteratively during the ground-state search until self consistency is reached, as in any other MF-based approach. By introduction of the $\alpha$-term in our MF-Hamiltonian, the model loses the particle-number conservation of the original 3D Hamiltonian. Thus, not only $\alpha$ but also the chemical potential $\mu$ has to be adjusted during the ground-state search.

At the inception of the iterative procedure $\alpha$ and $\mu$ must be guessed, however crudely. Then, we perform a DMRG-based ground-state search for this set of parameters, yielding a candidate for a ground state. Now, we need to check if the density is at the desired value and if $\alpha$ is consistent. First, we measure the density $\rho_{\text{current}}$ of the state we just calculated and compare it with the density $\rho_{\text{target}}$ we are targeting. If the condition

$$\frac{|\rho_{\text{current}} - \rho_{\text{target}}|}{|\rho_{\text{target}}|} < \varepsilon_\rho \quad \text{with} \quad \varepsilon_\rho \ll 1 \tag{20}$$

is fulfilled, we keep the chemical potential $\mu$ we plugged in, if not, a routine that involves interpolation and extrapolation is used to determine a new chemical potential which is applied from this point on. Second, we measure the value of the MF parameter $\alpha$ from the candidate state and check if it is converged via the condition

$$|\alpha_{\text{ini}} - \alpha_{\text{new}}| / |\alpha_{\text{ini}}| < \varepsilon_\alpha \quad \text{or} \quad |\alpha_{\text{new}}| < \varepsilon_\alpha . \tag{21}$$

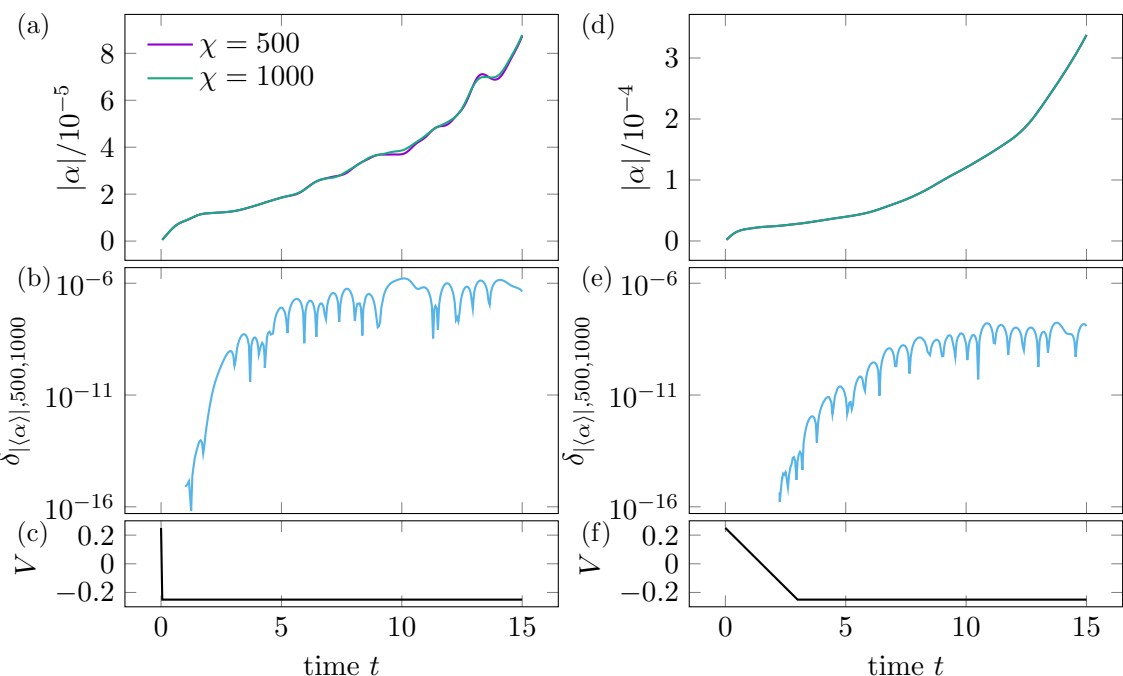

Figure 9: Comparing an instantaneous quench (left hand side) with a fast ramp in finite time (right hand side). Top row showing the time evolution of $|\alpha|$ for two different bond dimensions $\chi = 500$ and $\chi = 1000$ in a 12-site system, bottom row showing the difference between the two bond dimensions. We gain an accuracy of the order of $10^2$ via the fast ramp, as opposed to the instant quench.

If this condition is fulfilled, we keep $\alpha$, if not, we once again use a routine that involves extrapolation in order to find a new and better value for $\alpha$. Finally, we are either done if both conditions Eqs. (20) and (21) are fulfilled or we repeat the whole routine using now the new values we obtained for $\alpha$ and $\mu$ as a starting point.

A schematic of the self-consistent ground-state search is depicted in Fig. 8.

## B    Effect of the Time Window for the Ramp

In Sec. 4 it was mentioned that a ramp appeared to be numerically more stable than an instantaneous quench. For a more detailed explanation of this statement, we compare the accuracy of the data we measure for the MF parameter $|\alpha|$ for a quench and a ramp in Fig. 9.

Changing $V$ either through an instantaneous quench or through a fast continuous ramp, the latter of which we have used throughout the main text, we evolve our system up to times of $t_{\text{end}} = 15$. In both cases we compare the variance between the $\alpha$ data for two different bond dimensions $\chi$, as it was done in Sec. 4.2 as a check of accuracy. We find that difference is two orders of magnitude smaller for the ramp compared to the case of the instantaneous quench. This is why we chose to use ramps for all our calculations presented in this paper.

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
