# Peer review of "Transient superconductivity in three-dimensional Hubbard systems by combining matrix product states and self-consistent mean-field theory"

_SciPost Physics_

## Round 1 · Referee Report · Anonymous · 2022-10-24

Strengths

- Novel approach to simulating the Hubbard model in non-equilibrium
- technically sound and explained in proper detail

Weaknesses

- not clear enough on possible applications of the method and relation to experiments

Report

The authors propose a combined mean-field and matrix product state ansatz to simulate the time-evolution of an extended three-dimensional Hubbard model. The authors consider an attractive Hubbard interaction U at half-filling, where the equilibrium physics has already been intensely investigated. The manuscript offers a window into the dynamical properties of this system, and, hence, is highly relevant in the face of recent experiments on non-equilibrium superconductivity. The proposed simulation technique appears to be novel, and the technical skill of the authors allows them to perform the required simulations in a convincing manner. The authors convince especially with their accurate report of the convergence properties of their technique, something which is too often swept under the rug. A strong point of the manuscript is the self-contained and comprehensive explanation of the method.

However, I think there are various aspects that the authors need to address in further detail, before the manuscript can be considered for publication.

1) It should be clarified, how these simulations are relevant to experiments. Is the considered quench something that could be implemented in a solid-state environment, or in a putative quantum optics experiment? What sorts of experiments could be simulated with this technique? This is an important point and should be discussed in thorough detail to convince the reader that the proposed method is useful.

2) A discussion whether different non-equilibrium processes, e.g. some Floquet dynamics, could similarly be simulated with their method is missing. Would the method also be applicable in this case?

3) The authors should discuss in detail the potential shortcomings of the mean-field approximation. Since mean-field theory typically overestimates order, the authors should comment how this could alter their results in the context of these non-equilibrium simulations. A comparison to mean-field approaches vs. more-rigorous approaches in equilibrium might be a good guidance for this.

4) The figure captions are often a bit short require to go through the main text to understand the plot. As a reader, I would prefer to have more self-contained figure captions, to quickly grasp the contents of the manuscript.

I also have some minor comments:

a) please explain in more detail what is meant by the term "transient" superconductivity
b) On page 5 leading up to Eq. 7, it is not clear what is meant by E_{i,\alpha}. Please clarify this.

---

## Round 1 · Referee Report · Anonymous · 2022-12-23

Strengths

- Novel approach to non-equilibrium simulation embedded in a mean field
- Clearly written and presented

Weaknesses

- Lacks benchmarks to allow judgement of the trustworthiness
- Control of the results not convincing

Report

The authors present the extension of the approach of embedding 1D MPS simulations in a 3D mean field system to the non-equilibrium domain. The main advantage is that the approach allows to observe/simulate larger systems for longer real-time in an approximation best suited for highly anisotropic 3D systems. I think the manuscript/method presents a new approach with potential for more follow-up work in in- and out-of-equilibrium systems and as such should be considered for publication, however only after convincingly addressing the aspects below:

1) The work aims to test and benchmark the method itself. Unfortunately there is no comparison against any other method. As such it is hard judge, whether and how far one should trust the approach. While I understand that the comparison is hard due to the limited capability of other methods due to the dimensionality, I ask the authors to elaborate on this, test limiting cases, or at the very least propose potential routes to do so.

2) I think the the notion of 'dynamically induced' is not appropriate in the context of a MF approximation, as the symmetry has to be broken explicitly to allow for finite order parameters (\alpha_{ini} must be finite).

3) The density is assumed to be stable and the chemical potential is not determined self-consistently. However (1-\rho), while initially on a small scale, diverges rapidly beyond t~40, in particular for larger system sizes. This raises the question if one can trust any results beyond t~40. Figures 5 and 6 go well beyond that time scale, without showing the deviation of \rho from 1.

4) The dependence of the time evolution on \alpha_{ini} appears to be more influential than just being a scale, i.e., the evolutions for different \alpha_{ini} cannot be rescaled to collapse onto each other (c.f. Fig. 5). Self-consistent solutions in non-equilibrium MF do/should not depend on the initial MF parameters at the beginning of the self-consistency cycles. Here the situation seems to very different, especially beyond (again) the times beyond t~40. How helpful is the method if the results (time evolution) so heavily depends on \alpha_{ini}?

5) What are the used thresholds \varepsilon in the self-consistency cycle?

---

## Round 2 · Author Response

We thank the referees for their generally positive feedback. We note that the referees have highlighted the novelty of our approach with potential to yield future insight into many-body systems inside and out of equilibrium. Referee A has further commented favorably in the technical soundness of the present work and the clarity and self-contained presentation of our results. At the same time the referees have commented on some aspects of the present work and have asked for clarifications. In response, we hereby submit a revised manuscript and a diff-file which highlights the changes relative to the initially submitted manuscript. Furthermore, we respond to the queries of the referees in the present letter.

Respectfully,
Svenja Marten, Gunnar Bollmark, Thomas Köhler, Salvatore Manmana & Adrian Kantian

---

## Round 2 · List of Changes

- Substantially extended the Introduction and the Conclusion
- Added more details to figure captions
- Added more details to the explanation of the eigenenergies $E_{i,α}$
- Provide more information on the degree of overestimation of the order parameter and on the possibility for independent validation of the frameworks result
- Add the value of the precision parameter ε

You are currently on this page

Resubmission scipost_202208_00016v2 on 2 September 2023

---

## Editorial Decision

accepted_in_target_journal